Photoactivated disinfection procedure for denture stomatitis in diabetic rats

Zhang Xiao 1
Zhao Zirui 1
Zhang Ruiqi 1
Liu Juan 1
Guo Zhijiao 1
Hu Qiaoyu 1
Liu Na 2 liuna@hebmu.edu.cn
Liu Qing 1 liuqing@hebmu.edu.cn
1 Hebei Key Laboratory of Stomatology, Hebei Clinical Research Center for Oral Diseases, School and Hospital of Stomatology, Hebei Medical University , Hebei , China
2 Department of Preventive Dentistry, School and Hospital of Stomatology, Hebei Medical University , Hebei , China
Zhang Xin
Electronic publication date: 2024 Apr 30
Publication date: 2024
Volume: 12
Electronic Location ID: e17268
Received 2023 Nov 3; Accepted 2024 Mar 29
Copyright: © 2024 Zhang et al.
Copyright year: 2024
Copyright holder: Zhang et al.
License: This is an open access article distributed under the terms of the Creative Commons Attribution License, which permits unrestricted use, distribution, reproduction and adaptation in any medium and for any purpose provided that it is properly attributed. For attribution, the original author(s), title, publication source (PeerJ) and either DOI or URL of the article must be cited.
License URL: https://creativecommons.org/licenses/by/4.0/

Keywords: Candida albicans, Photoactivated disinfection, Diabetes mellitus, Denture stomatitis, IL-17, TNF-α

Funding: S&T Program of Hebei 20377799D Academic leader training program of Hebei Provincial Government 2018133206-2 Medical Science Research subject of Health Commission of Hebei Province 20191079 This work was supported by the S&T Program of Hebei (grant number 20377799D); the academic leader training program of the Hebei Provincial Government (grant number 2018133206-2); the Medical Science Research subject of Health Commission of Hebei Province (grant number 20191079). The funders had no role in study design, data collection and analysis, decision to publish, or preparation of the manuscript.

==============================
Objective

To study the efficacy of PADTM Plus-based photoactivated disinfection (PAD) for treating denture stomatitis (DS) in diabetic rats by establishing a diabetic rat DS model.

Methods

The diabetic rat DS model was developed by randomly selecting 2-month-old male Sprague-Dawley rats and dividing them into four groups. The palate and denture surfaces of rats in the PAD groups were incubated with 1 mg/mL toluidine blue O for 1 min each, followed by a 1-min exposure to 750-mW light-emitting diode light. The PAD-1 group received one radiation treatment, and the PAD-2 group received three radiation treatments over 5 days with a 1-day interval. The nystatin (NYS) group received treatment for 5 days with a suspension of NYS of 100,000 IU. The infection group did not receive any treatment. In each group, assessments included an inflammation score of the palate, tests for fungal load, histological evaluation, and immunohistochemical detection of interleukin-17 (IL-17) and tumor necrosis factor (TNF-α) conducted 1 and 7 days following the conclusion of treatment.

Results

One day after treatment, the fungal load on the palate and dentures, as well as the mean optical density values of IL-17 and TNF-α, were found to be greater in the infection group than in the other three treatment groups (P < 0.05). On the 7th day after treatment, these values were significantly higher in the infection group than in the PAD-2 and NYS groups (P < 0.05). Importantly, there were no differences between the infection and PAD-1 groups nor between the PAD-2 and NYS groups (P > 0.05).

Conclusions

PAD effectively reduced the fungal load and the expressions of IL-17 and TNF-α in the palate and denture of diabetic DS rats. The efficacy of multiple-light treatments was superior to that of single-light treatments and similar to that of NYS.

Introduction

Denture stomatitis (DS) is a common infection of the oral mucosa in denture wearers, and Candida albicans is the most significant etiological agent of DS (Sugio et al., 2020). Epidemiological studies have found that the incidence of DS in denture wearers ranges from 15% to 70% (Gendreau & Loewy, 2011). Traditional treatment involves antifungal drugs, but overuse has led to drug-resistant strains (Chen et al., 2021).

Diabetes mellitus (DM) is a common endocrine disorder that is escalating alarmingly. According to the International Diabetes Federation, 642 million adults will have diabetes worldwide by 2040, a significant increase from the 415 million reported in 2015 (Zimmet et al., 2016). Patients with DM are more susceptible to opportunistic infections (Gianchandani et al., 2020), including oral candidiasis, due to elevated serum glucose levels and decreased function of the cellular immune system (Khanna et al., 2021). High blood glucose levels in saliva are one of the main risk factors for oral Candida infection in patients with DM, with over 77% suffering from oral candidiasis (Soysa, Samaranayake & Ellepola, 2006). Studies indicate that patients with DM have a higher prevalence of C. albicans carriage than normal individuals and that their oral mucosa is more susceptible to fungal infections. The incidence of DS notably increases in patients with DM following the repair of removable dentures, further complicating the clinical management of DS (Javed et al., 2009). Manda et al. (2014) investigated the sensitivity of antifungal drugs to Candida in diabetic mice and found that high blood glucose levels reduce the activity of antifungal drugs. High blood glucose levels pose challenges in treating DS, necessitating the search for an effective clinical treatment method that does not induce drug resistance.

PADTM Plus-based photoactivated disinfection (PAD) is a novel therapy that selectively kills diseased cells or tissues through a photodynamic reaction generated by the interaction of light, photosensitizers (PS), and oxygen without damaging other normal tissues. Its main advantage is that microorganisms are less likely to develop resistance to reactive oxygen species (ROS) (Abdelkarim-Elafifi, Parada-Avendaño & Arnabat-Dominguez, 2021). In dentistry, PAD is widely used in caries, endodontics, periodontics, and oral clinical diagnosis and treatment (Haroon, Khabeer & Faridi, 2021). PAD has also proven effective in inactivating C. albicans. Most clinical studies have confirmed that PAD can effectively treat DS, but the treatment duration is long and requires multiple sessions. For example, Mima et al. (2012) incubated with 500 mg/L Photogem for 30 min, followed by 20 min of light irradiation. Alves et al. (2020) incubated with 200 mg/L Photodithazine for 20 min and irradiated for 4 min. Both studies were conducted thrice a week, totaling six sessions over 15 days. The prolonged treatment duration, along with extended mouth opening during oral treatment, may lead to complications such as temporomandibular joint disorders and excessive saliva secretion, potentially affecting the treatment outcome. Therefore, minimizing the oral operation time is crucial.

Fortunately, we have discovered the PAD technology, which effectively eliminates C. albicans within a short time. The technology uses a complementary pharmaceutical-grade 1 mg/mL toluidine blue O (TBO) solution and a 635-nm red light-emitting diode (LED) with an output power of 500 or 750 mW, along with an irradiation time of 1 or 2 min. The TBO solution is activated at this wavelength to produce ROS, which selectively kills microorganisms (Baltazar et al., 2013). In vitro studies have found that increasing the concentration of photosensitizers enhances the inhibitory effect on biofilms (Pinto et al., 2018). Previous research has shown that applying 1 mg/mL TBO for 1 min and 750-mW LED irradiation for 1 min had a good inactivation effect on C. albicans on the dorsal tongue of mice (Gu et al., 2022). Zhang et al. (2023) found that incubation with 1 mg/mL TBO for 1 min, followed by 1 min of 750-mW LED irradiation or 2 min of 500-mW LED irradiation, can inactivate over 99% of Candida in the mature mixed biofilms.

However, the efficacy of PAD on DS infection in diabetic rats compared to conventional antifungal drugs is unknown. Therefore, in this study, a 1 mg/mL TBO solution and a 750-mW LED red light for 1 min were selected for treatment, along with nystatin (NYS). We aimed to compare the therapeutic efficacy of PAD and NYS on DS infections in diabetic rats and explore potential differences in the efficacy of PAD with various durations of light irradiation in order to develop a rapid and efficient treatment method for the clinical treatment of DS.

Materials and Methods

Preparation of fungal suspension

C. albicans SC5314 was provided by Shijiazhuang Hera Biotechnology Co., Ltd (Hebei, China). The C. albicans strain stored at -80 °C was inoculated onto CHROMagarTM Candida chromogenic medium (Comagal Microbial Technology Co., Shanghai, China) and incubated at 37 °C for 24 h under a constant temperature incubator. After growth, a single colony was selected and re-inoculated onto a new Candida chromogenic medium and incubated at 37 °C for another 24 h. The activated single colony was then inoculated into 20 mL of yeast extract-peptone-dextrose medium broth medium (Kehua Jingwei Technology Co., Beijing, China) for amplification at 37 °C with shaking at 150 rpm overnight. The resulting fungal suspension was centrifuged at 4,000 rpm for 15 min in a high-speed centrifuge, and the supernatant was discarded. The fungal cells were rinsed in 10 mL of the PBS solution, the above mentioned centrifugation and rinsing steps were repeated thrice before discarding the supernatant to collect the fungal pellets. The pellets were then stored at 4 °C until further use. On the day of denture ligation, the spare fungal cells were diluted with PBS and counted on a blood cell count plate to 1 × 109 CFU/mL for inoculation into the rat palate.

Experimental animals and establishment of diabetes model in rats (Deeds et al., 2011; King, 2012)

Based on the animal attrition rate calculation of approximately 10% in the pre-experiment, 40 male SD rats (age: 2 months, weight: 300–350 g; Beijing Huafukang Biological Technology Co., Ltd., Beijing, China) (animal quality certificate number: 110322220101519632) were acclimatized at Bethune International Peace Hospital Animal Laboratory for 1 week; the environment was maintained at a constant temperature and humidity, and a paste diet (prepared by mixing ground standard pellet rat food with warm water) and water was provided to them ad libitum. This study was approved by the Ethics Committee of the Hospital of Stomatology, Hebei Medical University (Approval No: [2020]016).

A total of 40 rats were fasted for 12 h and then injected intraperitoneally with 60 mg/kg of 1% streptozotocin to induce diabetes. On the day of streptozotocin injection, the rats were fed with 5% sterile glucose water to prevent hypoglycemia, and normal sterile water was provided on the following day. After 72 h, the rats with a fasting blood glucose level of >16.7 mmol/L were considered to have successfully developed diabetes and were included in the experiment, while one rat that did not meet this criterion was excluded. Finally, 39 rats were found to have met the criterion and were analyzed in this study.

Denture fabrication (Yano et al., 2016)

The alginate impression material (Haijiya Medical Equipment Co., Beijing, China) was evenly spread on a tongue depressor and then placed in the rat mouth to obtain an impression of the palate (Fig. 1A). A corresponding gypsum (Hanhe Medical Equipment Co., Linyi, China) model was developed according to the impression (Fig. 1B). The denture was then prepared using light-cured acrylic resin (Huge Medical Equipment Co., Shanghai, China) on the gypsum model to approximately 3-mm thickness and used to cover the entire hard palate area (Fig. 1C). After polishing, it was placed in distilled water at 37 °C for 48 h to release any residual monomers. The prepared denture was then immersed in sterile distilled water and microwaved at 650 W for 3 min for sterilization.

Figure 1 Denture fabrication.

(A) Impression of the palate. (B) A corresponding gypsum model. (C) The light-cured acrylic denture.

Denture seeding (Lie Tobouti et al., 2016)

On the day of denture ligation, the prepared fungal suspension was diluted in PBS and counted using a hemocytometer to a concentration of 1 × 107 CFU/mL in RPMI-1640 medium for denture seeding. To allow C. albicans to adhere to the denture tissue surface, the denture was placed in a 6-well tissue culture plate, and each well contained 2 mL of 1 × 107 CFU/mL of C. albicans suspension in RPMI-1640 medium. The 6-well plate was placed in 37 °C water bath oscillation incubator and shaken at 75 rpm for 90 min. The denture was then gently immersed in 2 mL of PBS to remove any non-adherent fungal cells.

Denture ligation

The experimental rats were anesthetized via intraperitoneal injection of 0.6% (40 mg/kg) sodium pentobarbital. Two stainless steel ligatures (5-cm length, 0.2-mm diameter) were threaded between the first and second molars on either side of the maxilla and through the holes on both sides of the denture. The excess wire at the end was cut, and the tip was covered with a self-curing resin to protect the oral soft tissues of the rat. A fungal suspension (density 1 × 109 CFU/mL) was applied to the rat’s palate after denture ligation (Moraes et al., 2022). Preceding modeling experiments showed that a stable DS model in diabetic rats could be established after 3 weeks of ligation and denture seeding.

PAD treatment methods

A total of 36 rats were selected as the molded rats and assigned to four groups by using a random number table method (n = 9 per group). For the PAD-1 group, 1 mg/mL of the TBO solution (Denfotex, Edinburgh, UK) was applied to the palate and denture tissue surface with a small brush, incubated for 1 min, and then irradiated with 750-mW output power for 1 min (PADTM Plus instrument, Denfotex, UK; model: DX9001). The light source was systematically maneuvered to ensure that the entire palate and denture surface were irradiated. This treatment was conducted once a day for 1 day. For the PAD-2 group, the same treatment as for the PAD-1 group was administered, but with an interval of 1 day for irradiation, totaling three times within 5 days. The NYS group was treated with 100,000 IU of nystatin suspension applied to the palate and denture once a day for five consecutive days, while the infection group was not treated.

Efficacy observation and euthanasia

The rats in each group were anesthetized via intraperitoneal injection of 0.6% sodium pentobarbital at 40 mg/kg at 1 day and 7 days after the end of the treatment. The samples underwent two assessments: first, scoring for the extent of palatal inflammation, and second, measuring the fungal burden on both the palate and the denture tissue surface. Three rats from each group were randomly executed 1 day after the end of treatment (death by inhalation anesthesia with excessive isoflurane), and all remaining rats were executed 7 days after the end of treatment. The palate tissues were collected for histopathological examination and immunohistochemistry to detect the changes of interleukin-17 (IL-17) (Boorsen Biotechnology Co., Beijing, China) and tumor necrosis factor (TNF-α) (Bowan Biotechnology Co., Shanghai, China). Since preceding experiments have proven that the stable period of the model occurred at 3–6 weeks after denture seeding, the infection group was evaluated at the same time points as the PAD-1 and PAD-2 groups after treatment (at 1 day and 7 days, respectively), with the average data of the two-time points when compared to eliminate any bias caused by the time difference. The effectiveness of PAD therapy for diabetic rat DS was evaluated based on the results obtained from these evaluations.

Palate mucosal inflammation score. Newton’s method was applied to visually evaluate the palatal tissues, and the scores were assigned based on the severity of inflammation (Johnson et al., 2012). 0: no inflammation; 1: punctate erythema; 2: diffuse erythema and edema; 3: diffuse erythema/edema and papillary hyperplasia.

Palatal and denture tissue surface fungal burden measurement. The palate and denture of each group of rats were swabbed with sterile cotton for 1 min. The cotton swab was then placed into a centrifuge tube containing 1 mL of saline and shaken for 1 min. A 10-μL aliquot of each dilution was spread onto BIGGY agar culture medium and cultured at 25 °C for 48 h. The fungal colony count was measured in CFU/mL and analyzed statistically by taking the logarithm of the CFU number.

Histopathological examination

Hematoxylin and eosin (HE) staining: paraffin sections were baked at 60 °C, dewaxed and hydrated, followed by hematoxylin (Besso Biotechnology Co., Zhuhai, China; BA4097) and eosin (Besso Biotechnology Co., Zhuhai, China; BA4098) staining in sequence, dehydration, transparency and sealing.

Periodic acid-Schiff (PAS) staining: paraffin sections were baked at 60 °C, dewaxed and hydrated, followed by oxidization with periodate, then stained with Schiff’s reagent (Regan Biotechnology Co., Beijing, China; DG0005) and hematoxylin staining in sequence, dehydrated, transparent and sealed. Photographs were taken under microscope magnification of 400 times for observation.

Immunohistochemical (IHC) staining: paraffin sections were baked at 60 °C, dewaxed and hydrated, after which the antigen was repaired with citric acid antigen repair solution, heated to about 98 °C for 20 min; dropwise addition of appropriate amount of endogenous peroxidase blocker (SP-9000); dropwise addition of 20 μL of IL-17 antibody (1:100; bs-1183R), TNF-α antibody (1:50; AB3558), incubated at 37 °C for 2 h; dropwise addition of reaction enhancement solution (SP-9000) and incubated at 37 °C for 20 min; the enhanced enzyme-labeled goat anti-mouse/rabbit IgG polymer was added dropwise and incubated at 37 °C for 20 min, DAB was used to develop the color, hematoxylin was used for re-staining, washed and blown dry, and the slices were sealed with gum.

Microscopic observation, image acquisition, and analysis: Use a microscope (OLYMPUS BX63) to observe and collect images. Randomly select 3 non-repetitive fields (400x) from each slice, and use image analysis software IPP 6.0 (Madia Cybernetics Corp., Rockville, MD, USA) to measure the cumulative optical density (IOD) and area values of positive cells. Finally, the average optical density (IOD/area) value was calculated; the larger the value, the higher the relative expression of cytokines.

Statistical analysis

Data analysis was performed using SPSS25.0 software (IBM Corp., Armonk, NY, USA). The measurement data was tested for normality and homogeneity of variance tests. If normality and homogeneity of variance were met, two-factor ANOVA was used and group comparisons were performed using Student t-tests. Otherwise, the Friedman rank-sum test was applied and group comparisons were performed using the Kruskal-Wallis test. As the grade data (the palate mucosal inflammation score of the rats) used a constituent ratio, it was described and analyzed with the Friedman rank-sum test. P < 0.05 was considered to indicate statistical significance.

Results

Palate mucosal inflammation score

One day after treatment, the infection group exhibited apparent symptoms of palate swelling and redness, and the inflammation score was primarily 2 (Fig. 2A). In the three treatment groups, there was less palate inflammation, with scores between 0 and 1 (Figs. 2B–2D). The infection group’s palate inflammation persisted and worsened on day 7, with inflammation scores between 2 and 3 (Fig. 2E). Partial redness and edema of the palate were also observed in the PAD-1 group, with scores between 1 and 2 (Fig. 2F). The PAD-2 and NYS groups, with scores between 0 and 1, did not exhibit any discernible redness or swelling on the palate (Figs. 2G and 2H). One day after treatment, pairwise comparisons revealed that inflammation scores in the infection group were considerably higher than those of the three treatment groups (P < 0.05). Seven days after treatment, the inflammatory score in the infection group was noticeably greater than that in the PAD-2 group 7d after treatment (P < 0.05). The inflammation score in the PAD-1 group was significantly higher at 7 days after treatment than at 1 day after treatment (P < 0.05). However, there was no statistically significant difference in inflammation scores between the infection, PAD-2, and NYS groups at either 1 day or 7 days after treatment (P > 0.05) (Fig. 3). The inflammatory scores showed no significant difference between the infection, PAD-1, and NYS groups (P > 0.05, Table 1). Notably, the PAD-2 group demonstrated a statistically significant reduction in palatal inflammation, suggesting that its efficacy is potentially greater than that of the NYS group (Table 2). These findings suggest that only PAD-2 treatment can effectively reduce inflammation in this context.

Figure 2 Palate inflammation lesions of rats after treatment.

(A) Infection group, (B) PAD-1 group, (C) PAD-2 group, (D) NYS group of 1d after treatment. (E) Infection group, (F) PAD-1 group, (G) PAD-2 group, (H) NYS group of 7d after treatment.

Figure 3 Comparison of inflammation scores on the 1d and 7d after treatment.

*P < 0.05.

Table 1 Palate inflammation score in rats of 1d after treatment.

Group	N	Inflammation score	H value	P value	
0 points	1 point	2 points	3 points	
Infection group	6	0	1	4	1	12.580	<0.01	
(0%)	(16.67%)	(66.67%)	(16.67%)			
PAD-1 groupa	6	2	4	0	0			
(33.33%)	(66.67%)	(0%)	(0%)			
PAD-2 groupa	6	3	3	0	0			
(50%)	(50%)	(0%)	(0%)			
NYS groupa	6	2	4	0	0			
(33.33%)	(66.67%)	(0%)	(0%)			
Note:

a Compared with infection group, P < 0.05.

Table 2 Palate inflammation score in rats of 7d after treatment.

Group	N	Inflammation score	H value	P value	
0 points	1 point	2 point	3 points	
Infection group	6	0	0	4	2	9.387	<0.05	
(0%)	(0%)	(66.67%)	(33.33%)			
PAD-1 group	6	1	2	2	1			
(16.67%)	(33.33%)	(33.33%)	(16.67%)			
PAD-2 groupa	6	3	2	1	0			
(50%)	(33.33%)	(16.67%)	(0%)			
NYS group	6	2	2	2	0			
(33.33%)	(33.33%)	(33.33%)	(0%)			
Note:

a Compared with infection group, P < 0.05.

Fungal burden analysis

The fungal burden in the palate and dentures of the PAD-1 group was significantly higher at 7 days after treatment than at 1 day after treatment (P < 0.05). However, there were no significant differences in the palate and denture fungal load between the infection, PAD-2, and NYS groups at either 1 day or 7 days after treatment (P > 0.05; Fig. 4).

Figure 4 Comparison of palate and denture fungal burden at 1d and 7d after treatment.

(A) Palate fungal burden. (B) denture fungal burden. *P < 0.05, **P < 0.01. (N = 6).

When comparing pairwise, the fungal load in the palate of the three treatment groups was significantly lower than that in the infection group 1day after treatment, with statistical significance (P < 0.05; Fig. 5A). There were no significant differences in the palate fungal load between the three treatment groups (P > 0.05). On day 7 after treatment, the fungal load in the palate of the infection and PAD-1 groups was significantly higher than that of the PAD-2 and NYS groups (Fig. 5B, P < 0.05). There was no statistically significant difference between the PAD-1 and infection groups, as well as between the PAD-2 and NYS groups (P > 0.05; Fig. 5B). These results suggest that, when used efficiently, PAD can kill C. albicans, achieving a level similar to that of NYS.

Figure 5 Palate fungal burden of each group after treatment.

(A) 1d after treatment. (B) 7d after treatment. ***P < 0.001, ****P < 0.0001. (N = 6).

When comparing pairwise, the denture fungal burden in the infection group was significantly higher than that in the three treatment groups 1 day after treatment, with statistical significance (Figs. 4B and 6; P < 0.05). There were no significant differences in denture fungal colonization between the three treatment groups (P > 0.05). However, 7 days after treatment, the denture fungal load in the infection group was significantly higher than that in the PAD-2 and NYS groups (Fig. 4B; P < 0.05). Interestingly, the denture fungal burden in the PAD-1 group was also significantly higher between days 1 and 7 (Fig. 4B) than that in the PAD-2 group, with significant differences (Fig. 6B; P < 0.05). However, there were no significant differences in the denture fungal load between the PAD-1 and infection groups; the PAD-2 and NYS groups; and the PAD-1 and NYS groups (P > 0.05; Fig. 6B).

Figure 6 Denture fungal burden of each group after treatment.

(A) 1d after treatment. (B) 7d after treatment. *P < 0.05, **P < 0.01. (N = 6).

Palate tissue histology

As observed in the HE staining of the tissue in Fig. 7, one day after colonization, the infection group had notable thickening of the epithelial layer with papillary projections (Fig. 7A), while the epithelial layer structure appeared more homogenous in the three treatment groups (Figs. 7B–7D). In contrast, 7 days after treatment, the epithelial tissue in the infection group showed significant and abnormal proliferation, while partial proliferation was observed in the epithelium of the PAD-1 group (Figs. 7E and 7F). The epithelial structure appeared more normal in the PAD-2 and NYS groups, with no significant differences in epithelial structure among these treatment groups (Figs. 7G and 7H).

Figure 7 HE staining of palate mucosa of each group after treatment.

(A) Infection group, (B) PAD-1 group, (C) PAD-2 group, (D) NYS group of 1d after treatment. (E) Infection group, (F) PAD-1 group, (G) PAD-2 group, (H) NYS group of 7d after treatment. The epithelial layer shown by the arrows in (A) and (E–H) is significantly thickened, with papillary protrusions visible. The epithelial layer structure shown by the arrow in (B–D) is relatively normal.

The presence of fungal colonization was analyzed with PAS staining of the mucosa. One day after treatment, PAS staining revealed that a number of C. albicans yeast cells adhered to the tissue surface in the infection group and that some hyphae invaded the superficial epithelium. The number of C. albicans in the three treatment groups was less evident compared to that in the infection group, and occasional C. albicans yeast cells were observed to adhere to the mucosal surface but without hyphae invasion (Figs. 8A–8D). There were no significant differences in fungal load among the treatment groups. Seven days after treatment, a large number of C. albicans yeast cells adhered to the mucosal surface in the infection group, with hyphae invading the superficial part of the epithelium (Fig. 8E). The PAD-1 group showed an intermediate number of C. albicans yeast cells adhered to the mucosal surface and partial hyphae invasion of the epithelium, but less than that observed in the infection group (Fig. 8F). Occasionally, a few C. albicans yeast cells were observed on the mucosal surface in the PAD-2 and NYS groups (Figs. 8G and 8H), supporting the previous findings.

Figure 8 PAS staining of palate mucosa of each group after treatment.

(A) Infection group, (B) PAD-1 group, (C) PAD-2 group, (D) NYS group of 1d after treatment. (E) Infection group, (F) PAD-1 group, (G) PAD-2 group, (H) NYS group of 7d after treatment. Candida hyphae and spores shown by arrows in (A–H).

The effect of PAD on IL-17 and TNF-α expressions in the palate mucosa of diabetic DS rats

IL-17 is a secretory protein that can be detected through immunohistochemistry as a brownish-yellow pattern in the cytoplasm and intercellular space of various layers of epithelial cells in normal tissues. One day after treatment, the infection group exhibited strong positive expression in various layers of epithelial cells, stroma, and vascular endothelial cells (Fig. 9A). The treatment groups were weakly positive in epithelial cells, stroma, and vascular endothelial cells, with lighter staining compared to the infection group (Figs. 9B–9D). After 7 days of treatment, the infection and PAD-1 groups showed strong positive expression in epithelial cells, stroma, and vascular endothelial cells (Figs. 9E and 9F). In comparison, the PAD-2 and NYS groups exhibited a weaker expression pattern in these cell types, with lighter staining than the infection and PAD-1 groups (Figs. 9G and 9H).

Figure 9 Expression of IL-17 in the palate mucosa of each group after treatment.

(A) Infection group, (B) PAD-1 group, (C) PAD-2 group, (D) NYS group of 1d after treatment. (E) Infection group, (F) PAD-1 group, (G) PAD-2 group, (H) NYS group of 7d after treatment.

On day 1 after treatment, the average optical density value of the infection group was significantly higher than that of the treatment groups (Table 3, a super index P < 0.05). The average optical density value of the PAD-1 group was also slightly higher than that of the PAD-2 and NYS groups (P < 0.05). There was no significant difference in the optical density value between the PAD-2 and NYS groups (Table 3, b super index P > 0.05). Seven days after treatment, the average optical density values of the infection group and PAD-1 group were significantly higher than those of the PAD-2 and NYS groups (P < 0.05). There was no significant difference in the optical density value between the infection and PAD-1 groups, as well as between the PAD-2 and NYS groups (P > 0.05) (Table 3).

Table 3 Average optical density values of IL-17 of each group after treatment.

Group	N	IL-17 average optical density values
χ ± s	t value	P value	
1d	7d	
Infection group	3	0.1051 ± 0.0052	0.1201 ± 0.0105	−1.722	0.227	
PAD-1 group	3	0.0575 ± 0.0040a	0.1158 ± 0.0251	−4.525	0.046	
PAD-2 group	3	0.0414 ± 0.0020ab	0.0630 ± 0.0056ab	−7.291	0.018	
NYS group	3	0.0416 ± 0.0057ab	0.0680 ± 0.0055ab	−4.719	0.042	
Notes:

a Compared with infection group, P < 0.05.

b Compared with PAD-1 group, P < 0.05.

TNF-α can also be expressed in normal tissues, mainly in the cytoplasm of cells across the epithelial layer. It appears as yellow-stained particles in the cytoplasm via immunohistochemistry. A strong positive expression of TNF-α was observed in the epithelial and vascular endothelial cells in the infection group. One day after treatment, a weak positive expression of TNF-α was detected in the epithelial and vascular endothelial cells of the treatment groups, with lighter staining compared to the infection group. Seven days after treatment, a strong positive expression of TNF-α was observed in the surface and basal layers of the epithelium and vascular endothelial cells in the PAD-1 group, and a positive expression was observed in the granular and spinous layers. A weak positive expression of TNF-α was observed in the epithelial and vascular endothelial cells of the PAD-2 and NYS groups, with lighter staining than the infection and PAD-1 groups (Fig. 10).

Figure 10 Expression of TNF-α in the palate mucosa of each group after treatment.

(A) Infection group, (B) PAD-1 group, (C) PAD-2 group, (D) NYS group of 1d after treatment. (E) Infection group, (F) PAD-1 group, (G) PAD-2 group, (H) NYS group of 7d after treatment.

When comparing pairwise, the average optical density value in the infection group was significantly higher than that in the treatment groups 1 day after treatment (P < 0.05). Conversely, there was no significant difference in the optical density value among the treatment groups (P > 0.05). Seven days after treatment, the average optical density values in the infection and PAD-1 groups were significantly higher than those in the PAD-2 and NYS groups (P < 0.05). There was no significant difference in the optical density values between the infection and PAD-1 groups, as well as between the PAD-2 and NYS groups (P > 0.05) (Table 4).

Table 4 Average optical density values of TNF-α of each group after treatment.

Group	N	TNF-α average optical density values
χ ± s	t value	P value	
1d	7d	
Infection group	3	0.0723 ± 0.0004	0.0756 ± 0.0006	−6.043	0.026	
PAD-1 group	3	0.0653 ± 0.0018a	0.0732 ± 0.0031	−10.678	0.009	
PAD-2 group	3	0.0640 ± 0.0002a	0.0659 ± 0.0011ab	−2.876	0.103	
NYS group	3	0.0665 ± 0.0037a	0.0672 ± 0.0025ab	−1.958	0.189	
Notes:

a Compared with infection group, P < 0.05.

b Compared with PAD-1 group, P < 0.05.

Discussion

PAD is a noninvasive treatment method that was initially developed in dermatology and later applied to cancer treatment (Zhao et al., 2012). In recent years, it has been adopted in the treatment of oral mucosal diseases. Its mechanism of action is based on the interaction between a specific wavelength light source and PS in the presence of oxygen, inducing specific cell damage. PS absorbs photons from a light source of a specific wavelength, causing the energy to transition from a lower ground state to a higher excited singlet state. The excited singlet state may decay over time with laser irradiation or transition to the triplet state. This excited triplet state can undergo two reactions: In the Type I reaction, electrons or hydrogen molecules are stripped from the substrate to generate highly active free radicals, which then react with endogenous oxygen molecules to produce ROS such as hydrogen peroxide, hydroxyl radicals, and superoxide, leading to massive cellular damage. In the Type II reaction, the PS reaches the triplet excited state and reacts with oxygen molecules that exist in the target cells, such as gram-negative and positive bacteria, producing highly reactive oxygen or singlet oxygen molecules. Oxidative damage affects the target cells’ plasma membrane, including proteins, lipids, and DNA, leading to cell death without affecting host cell activity (Abdelkarim-Elafifi, Parada-Avendaño & Arnabat-Dominguez, 2021). Notably, the Type II reaction is commonly used for anti-infective treatment (Konopka & Goslinski, 2007).

Recent years have seen a significant shift in research on PAD for DS, with a greater emphasis on clinical trials and a reduced focus on fundamental investigations. As clinical research is inconvenient for relevant studies on pathology and inflammatory factors, this study uses PAD technology to treat DS in diabetic rats. The study results demonstrate that PAD alleviates palatal inflammation in rats, with all treatment groups exhibiting good fungicidal effects against C. albicans 1 day after treatment. The PAD-2 group had the lowest fungal burden in palatal and denture samples compared to the infection group. Consequently, the experimental outcomes, aimed at reducing inflammation and sterilizing the palate, align with the findings of some previous clinical research (Afroozi et al., 2019; Alrabiah et al., 2019; Mima et al., 2011; de Senna et al., 2018).

Owing to prior research indicating the possibility of disease recurrence during the follow-up phase after DS treatment, the present study continued observation until 7 days post-treatment (Alves et al., 2020; Mima et al., 2011). After 7 days of treatment, the palatal and denture fungal burden in the PAD-1 group showed no statistically significant difference compared to the infection group, suggesting the onset of DS recurrence. However, both the PAD-2 and NYS groups exhibited significantly lower fungal burdens on the palate and dentures than infection and PAD-1 groups at 7 days post-treatment, although slightly higher than at 1-day post-treatment. This observation might be attributed to the acrylic resin composition of the dentures, which can function as a reservoir for microorganisms and pose a risk of patient re-infection. Disinfection of dentures is crucial for DS treatment (Wezgowiec et al., 2022a, 2022b). Replacing old dentures is often crucial for full DS resolution, particularly when the dentures are significantly aged. Regular denture renewal plays a vital role in effectively managing and preventing this infection.

There was no statistically significant fungal burden on the palate and denture in the PAD-2 and NYS groups at days 1 and 7 post-treatment, indicating that the efficacy of multiple photodynamic therapy was similar to that of traditional treatment with NYS, consistent with the experimental results of Scwingel et al. (2012).

Notably, PAD stands out for its shorter treatment duration, markedly reducing both the time and frequency of treatment for patients and offering greater convenience. Additionally, PAD’s low likelihood of developing drug resistance, coupled with its biocompatibility and minimal side effects, positions it as a highly favorable option. Thus, PAD emerges as a promising approach for DS treatment.

DM is a chronic metabolic disease characterized by hyperglycemia and influenced by multiple factors such as lifestyle, genetics, and environment, with a slightly higher prevalence in women. Hyperglycemia weakens the immune system, increasing the risk of infection in patients with DM. Fungal infection is common in DM (Khanna et al., 2021). Numerous risk factors such as age, gender, nutrition, oral hygiene, smoking, and dentures make patients with DM more susceptible to oral candidiasis due to high salivary glucose levels, low salivary secretion, impaired chemotaxis, and phagocytosis defects caused by polymorphonuclear leukocyte deficiency (Tabesh, Mahmood & Sirous, 2023; Mohammadi, Javaheri & Nekoeian, 2016). An in vitro study has shown that blood glucose levels in normal individuals (0.1%) are sufficient to enhance the expressions of mycelium-associated genes (Buu & Chen, 2014). Thus, higher glucose levels in patients with DM may induce C. albicans mycelium formation and promote DS development.

In the treatment of fungal infections, in addition to the direct effect of drugs, the body’s anti-infective immunity is also crucial. In adaptive immunity, Th17 cells and their secretion of IL-17 are important mechanisms for regulating fungal immunity and protecting the body from fungal infection. Currently, few studies have documented the immunological response during DS. IL-17, a characteristic cytokine secreted by Th17 cells, is a potent multi-effect pro-inflammatory factor mainly secreted by activated CD4+ T lymphocytes. It is a mediator of various immune, autoimmune, and inflammatory disorders, including DM (Roohi et al., 2014). Th17 cells secrete the pro-inflammatory cytokine IL-17, which, in the presence of local inflammation, binds to receptors expressed on the mucosal epithelial cells in the oral cavity. This leads to the release of relevant chemokines and the stimulation of massive secretions of inflammatory cytokines, either promoting or exacerbating the inflammatory response (Lee et al., 2015). Research has shown that IL-17 from the same cell can not only induce neutrophils to eliminate pathogens-playing a protective role-but can also induce excessive inflammatory reactions that damage the tissue (Matsuzaki & Umemura, 2018). TNF-α secreted by various immune cells such as macrophages, monocytes, neutrophils, and CD4+T cells, is a pro-inflammatory cytokine that participates in inflammatory and immune responses. It can also synergistically regulate the production of other cytokines. IL-17 and TNF-α play a synergistic role in the pathogenesis of diseases such as psoriasis. IL-17 can induce macrophages to secrete IL-1β and TNF-α (McGeachy, Cua & Gaffen, 2019). Although IL-17 is crucial in combating microbial infections by triggering the induction of inflammatory cytokines and chemokines, it is also involved in the development of many inflammatory diseases, such as autoimmune and metabolic disorders and cancer. It induces inflammation alone or in combination with TNF-α, which aggregates and fibrillates immune cells (Robert & Miossec, 2017).

PAD has a significant inhibitory effect on C. albicans in vitro (Zhang et al., 2023). However, it remains unclear whether PAD can regulate antifungal immunity in vivo, especially through the Th17/IL-17 immunoinflammatory pathway. Therefore, we established a rat model of DS in this experiment and investigated the immunological mechanism of PAD in DS treatment by observing the expression of inflammatory cytokines IL-17 and TNF-α related to the Th17/IL-17 pathway.

In recent years, the identification of genetic defects in the Th17/IL-17 axis in both mice and humans has highlighted the importance of this pathway in controlling Candida infection. Patients with specific defects in IL-17 immunity, such as mutations affecting IL-17 production or receptor function, have shown increased susceptibility to chronic cutaneous or mucosal candidiasis. Before identifying Th17 cells, the immune response mediated by IL-12 and Th1 cells was considered to play a major protective role in mucosal candidiasis (Conti & Gaffen, 2010). Schönherr et al. (2017) developed a mouse oral candidiasis model and found that while the neutrophil recruitment and inflammatory response triggered by IL-17 varied markedly across different Candida species, the essential role of IL-17 in establishing mucosal immunity against fungal infections remained consistent across all examined fungal species. Conti et al. (2009) demonstrated that Th17-deficient mice were extremely sensitive to oral pharyngeal candidiasis. Saijo et al. (2010) found that IL-17A and IL-17RA knockout mice are more susceptible to Candida infection compared to wild-type mice in systemic Candida infections. In previous pre-experiments, attempts to induce Candida infection in normal rats were unsuccessful, potentially due to the presence of the Th17/IL-17 axis. In the present study, semi-quantitative analysis of IL-17 and TNF-α in the post-treatment groups revealed significantly higher optical density values for both factors in the infection group at 1-day post-treatment. Additionally, these values for both factors were significantly higher in the infection and PAD-1 groups than in the PAD-2 and NYS groups at 7 days post-treatment (P < 0.05). This suggests a potential synergistic action between PAD and the Th17/IL-17 axis in controlling Candida infections. IL-17 and TNF-α may be involved in the immunomodulation of DS, and the effects of PAD on immune cells and whether the immune responses it stimulates contribute to DS treatment need to be further explored.

Although the experimental results showed that PAD technology achieved better results in the treatment of DS in diabetic rats and shortened treatment time, there are some limitations in this study. DM was not controlled during the experiment, and the impact of DM on the treatment effect and long-term prognosis remains unexplored. Additionally, a small number of C. albicans remained after the treatment, suggesting that future experiments could consider controlling for DM, increasing the number of irradiations, or combining with antifungal drugs to develop a more convenient and effective treatment protocol.

In summary, we used a 1 mg/mL TBO solution, incubated for 1 min, together with 750 mW output power LED light source illumination for 1 min to treat DS in diabetic rats and evaluated its therapeutic effect. The experimental results showed that PAD for treating DS greatly reduced the burden of C. albicans in the palate and denture of the rats, improved inflammation symptoms in the palate, and decreased IL-17 and TNF-17 in the palate tissues of diabetic rats with DS. We found that multiple-light treatments are better than single-light treatments, and no adverse reactions were observed in terms of safety. Compared with previous studies, the operation time was markedly shorter, offering practicality and convenience for oral treatment.

Supplemental Information

Supplemental Information 1 Raw data.

Supplemental Information 2 The ARRIVE guidelines 2.0: author checklist.

Additional Information and Declarations

Competing Interests

Author Contributions

Animal Ethics

Data Availability

The authors declare that they have no competing interests.

Xiao Zhang conceived and designed the experiments, performed the experiments, analyzed the data, prepared figures and/or tables, and approved the final draft.

Zirui Zhao performed the experiments, authored or reviewed drafts of the article, and approved the final draft.

Ruiqi Zhang performed the experiments, authored or reviewed drafts of the article, and approved the final draft.

Juan Liu performed the experiments, authored or reviewed drafts of the article, and approved the final draft.

Zhijiao Guo performed the experiments, authored or reviewed drafts of the article, and approved the final draft.

Qiaoyu Hu performed the experiments, authored or reviewed drafts of the article, and approved the final draft.

Na Liu conceived and designed the experiments, authored or reviewed drafts of the article, and approved the final draft.

Qing Liu conceived and designed the experiments, authored or reviewed drafts of the article, and approved the final draft.

The following information was supplied relating to ethical approvals (i.e., approving body and any reference numbers):

The use of animals in this study was approved by the Ethics Committee of the Hospital of Stomatology, Hebei Medical University, approval number: [2020]016.

The following information was supplied regarding data availability:

The raw data are available in the Supplemental File.

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
