# Peer review of "Photoactivated disinfection procedure for denture stomatitis in diabetic rats"

_PeerJ, doi:10.7717/peerj.17268_

## Round 0.1 · original submission · Major Revisions

· Academic Editor

Major Revisions

Please revise the manuscript carefully and answer the reviewer's questions.

Reviewers have suggested that you cite specific references. You are welcome to add it/them if you believe they are relevant. However, you are not required to include these citations, and if you do not include them, this will not influence my decision.

Reviewer 1 ·

Basic reporting

The topic is novel and interesting. The study reporting is average.
I found a few major and minor flaws:
1. Authors have to define why this study is novel and important before the aim of the study at the end of Introduction (please rationale the study).
2. Authors have to define clear aim of the study at the end of Introduction.
3. Authors have to present that photoactivated disinfection is used in different dental specializations based on the following reliable literature (Authors can add a paragraph within Introduction):

-Haroon S, Khabeer A, Faridi MA. Light-activated disinfection in endodontics: A comprehensive review. Dent Med Probl. 2021;58(3):411–418. doi:10.17219/dmp/133892
-Wezgowiec J, Wieczynska A, Wieckiewicz M, Czarny A, Malysa A, Seweryn P, Zietek M, Paradowska-Stolarz A. Evaluation of Antimicrobial Efficacy of UVC Radiation, Gaseous Ozone, and Liquid Chemicals Used for Disinfection of Silicone Dental Impression Materials. Materials (Basel). 2022 Mar 31;15(7):2553. doi: 10.3390/ma15072553.
-Wezgowiec J, Paradowska-Stolarz A, Malysa A, Orzeszek S, Seweryn P, Wieckiewicz M. Effects of Various Disinfection Methods on the Material Properties of Silicone Dental Impressions of Different Types and Viscosities. Int J Mol Sci. 2022 Sep 17;23(18):10859. doi: 10.3390/ijms231810859.
4. Please describe a study limitations at the end of Discussion.
5. Please define clear conclusions at the end of manuscript body.

Experimental design

1. Authors have to provide details about keeping rats and their food. In what laboratory were the rats kept?
2. Authors have to provide details regarding materials, devices, and manufactures used in dentures fabrication. I would like to read brand names and manufacturers, country of origin of each material and device used in the study.
3. Authors should discuss the following latest and outstanding articles related to candida infection, diabetes, and acrylic resin which are important for the discipline:

- Darwazeh A, Al-Shorman H, Mrayan B. Effect of statin therapy on oral Candida carriage in hyperlipidemia patients: A pioneer study. Dent Med Probl. 2022;59(1):93–97. doi:10.17219/dmp/142641
- Altaie SF. Tribological, microhardness and color stability properties of a heat-cured acrylic resin denture base after reinforcement with different types of nanofiller particles. Dent Med Probl. 2023;60(2):295–302. doi:10.17219/dmp/137611
- Tabesh A, Mahmood M, Sirous S. Oral health-related quality of life and xerostomia in type 2 diabetic patients. Dent Med Probl. 2023;60(2):227–231. doi:10.17219/dmp/147754
4. This branch of oral sciences is growing very fast. Please do not cite articles older than 10 years because mostly they are outdated.
5. Please add a legend of used abbreviation below each table and figure.

Validity of the findings

The validity of findings is high and clinically useful. I don't have further comments.

Additional comments

The manuscript will be ready to be publish after major revision.

·

Basic reporting

Perhaps an area for improvement in this work lies in the language used. Some phrases in the introduction need better contextualization and a stronger bibliographic foundation. At times, certain phrases are not entirely clear, or there are distortions that obscure the essence of the ideas the authors intend to convey. While the context becomes understandable when reading the rest of the document, especially for those familiar with candidiasis, it might pose a challenge for readers from different fields outside clinical mycology. In the field of stomatology, I find the work to be quite clear, though there are some loose ends regarding the immune response, particularly in the context of diabetes. In this regard, I have several observations and concerns that I will detail further on.
The authors' work focuses on a very interesting topic: the use of phototherapy for the treatment of dental stomatitis caused by Candida albicans, the primary opportunistic pathogen of these infections worldwide. Despite the novelty of the subject, I believe there are some questions that would benefit from the echo of previous research. For instance, I suggest some references that could enrich the authors' work are:
• Gianchandani, R., Esfandiari, N., Ang, L., Iyengar, J., Knotts, S., Choksi, P., & Pop-Busui, R. (2020). Managing Hyperglycemia in the COVID-19 Inflammatory Storm.Diabetes, 69, 2048 -2053.https://doi.org/10.2337/dbi20-0022.
• Khanna, M., Challa, S., Kabeil, A., Inyang, B., Gondal, F., Abah, G., Dhandapani, M., Manne, M., & Mohammed, L. (2021). Risk of Mucormycosis in Diabetes Mellitus: A Systematic Review.Cureus, 13.https://doi.org/10.7759/cureus.18827
• Baltazar, L., Soares, B., Carneiro, H., Ávila, T., Gouveia, L., Souza, D., Ferreira, M., Pinotti, M., Santos, D., & Cisalpino, P. (2013). Photodynamic inhibition of Trichophyton rubrum: in vitro activity and the role of oxidative and nitrosative bursts in fungal death..The Journal of antimicrobial chemotherapy, 68 2, 354-61 .https://doi.org/10.1093/jac/dks414.

I greatly appreciate the authors sharing raw data for analysis in SPSS, Stata, or R. I find the statistical bases well-justified and selected, though details on the verification of data parametricity for normality adjustment are missing. The structure and quality of both the tables and figures are appropriate; I only have a few recommendations for figures 3, 7, and 8. I believe Figure 3 could be substantially improved.
The results provided are robust enough to determine the efficiency of treatment with phototherapy. However, I see a significant issue with the experimental approach: the lack of a control group of non-diabetic rats subjected to the same experimental conditions is notable. Such a group would have made the experiment more comprehensive and elucidated differences in colonization between normal hosts and those with a metabolic condition as concerning as diabetes mellitus. Including these experiments would have provided support for some of the assertions referenced in the discussion and introduction, specifically in the paragraphs covering lines 39-55.
In the attached document, I include comments that, if heeded, would help improve the understanding of what is displayed in the figures and tables.

Experimental design

Regarding the scope of the article, I believe that it does meet the editorial criteria to be included in PeerJ, however, the comments of the peer reviewers should be addressed and their and my questions answered throughout the document.
While the research question is well-formulated from the perspective of the efficacy of phototherapy compared to traditional nystatin treatment, the study lacked an exploration of the influence of metabolic conditions in the rats. The approach and results achieved shed light on optimal conditions for eliminating the infectious agent in the dental stomatitis model. However, the potential synergy of combining both treatments, phototherapy and nystatin, was not explored in depth, although the authors do mention this towards the end of their discussion. I reiterate that including a control group of non-diabetic rats would have been highly beneficial, aiding in understanding the differences in Candida albicans colonization dynamics under normal and diabetic conditions. In this sense, I feel the experiment is incomplete and would have better bridged the knowledge gap by including these healthy rats for comparative analysis, both in terms of colonization and cytokine differences.
I also appreciate the authors' inclusion of the bioethics committee approval letter from their institution and look forward to discovering the findings in the third phase of their project.
Furthermore, I have several concerns regarding the methodology. Some details seem to be missing in the preparation of the fungal suspensions (lines 92-94) and the number of rats that met the criteria; I would prefer this to be mentioned from line 113 rather than later. The volume of conidial suspension used for denture colonization (line 127) is also unclear; while the concentration of the suspension is mentioned, this alone does not suffice to gauge the quantity of fungi used. Lastly, it seems imperative to add a section titled 'Histopathological Examination.' This section should detail the microscope used and, most importantly, the methodologies for all stains: hematoxylin-eosin, Periodic Acid-Schiff, and more detailed information on the antibodies used, including catalog numbers and other important details for detecting both TNF-α and IL-17. It is also important to mention here the software used for quantifying immunohistochemical signals – was it ImageJ, for instance? If so, it should be noted. This is crucial for enabling other researchers to replicate the experiments.

Validity of the findings

In this section, I reiterate that the methodology must be explained in more detail so that another researcher can reproduce the comparisons made by the authors.
In the arguments presented in the first section of the results, I would appreciate it if the authors could provide clearer and more emphatic clarification on whether there were differences in the scores observed on day 1 and day 7 among the treated groups. Another concern that arises for me is that the images selected for the tables do not seem entirely comparable, as those chosen for the groups of infected rats display a larger section of the dermis and dermal papillae than the epidermis, which is more predominant in the treated groups. Therefore, I would encourage the authors to share more data and photographs from other fields to observe the complete pattern across all treatment conditions and infected rats.
In the discussion of the results, I recommend a more elaborate transition to assist the reader in understanding the rationale for focusing on the detection of the cytokines TNF-alpha and IL-17 (line 348), particularly in the context of the disease, diabetes. In this section, the concept of the ‘white’ phenotype of Candida albicans is also introduced, but it seems out of context, as the sexual cycle of the pathogen is not referred to at any other point in the article. Clarification on this aspect (line 361) is needed.
Finally, while the article offers statistically robust results, a more in-depth discussion on the significance of these findings within the Th17/IL-17 axis in diabetic rats is required.

---

## Round 0.2 · Minor Revisions

· Academic Editor

Minor Revisions

Further revision of the manuscript is required.

Reviewer 1 ·

Basic reporting

The manuscript has been revised correctly. I don't have further comments.

Experimental design

The manuscript has been revised correctly. I don't have further comments.

Validity of the findings

The manuscript has been revised correctly. I don't have further comments.

Additional comments

The manuscript has been revised correctly. I don't have further comments.

·

Basic reporting

Thank you very much for incorporating our suggestions and recommendations. Overall, I am very pleased with the authors' responses. However, while relevant information has been added and the manuscript has been restructured for improved clarity, there remain areas for improvement:
I appreciate the enhanced writing quality and the improved handling of the English language. I am also grateful for the inclusion of the suggested references. In this context, I would like to request the addition of some references to support the statements made in the following manuscript lines: Lines 37-38 and 156-158.
In the latest submission of the document, I have attached a document with tracked changes where I suggest the inclusion of these references. In this series of files, I have also included some suggestions to improve the aesthetics of your graphics. To standardize the tables, I have recommended bracketing all letters that denote statistical differences in Tables 1, 2, and 3 to ensure uniformity.

Experimental design

The attention given to addressing the reviewers' comments has satisfactorily met the criteria required to support the hypotheses, and the discussion has been well-rounded with your ideas. My comments are limited to the technical aspects introduced in your latest submission. I believe that the writing in lines 199-226 could be improved, and some details regarding catalog numbers should be included to enable replication in the laboratory. Specifically, I recommend including catalog numbers for materials, kits, and antibodies used in the histopathological examination. It is also important to include the name of the developer of the IPP 6.0 software and its country of origin, similarly to how SPSS was specified. Thank you for detailing the statistical tests employed.

Validity of the findings

Thank you to the authors for providing the raw data. Your discussions on the IL-17 axis, candidiasis under the metabolic condition of diabetes mellitus are very interesting and valuable in devising strategies to control candidal stomatitis with an approach that avoids the use of antifungals, with the aim of preventing the emergence of resistance. It was also valuable to include in the discussion that DM was not controlled during the experiment.

---

## Round 0.3 · accepted · Accept

· Academic Editor

Accept

The revisions suggested by the reviewers have been largely refined by the authors. I reviewed the manuscript and determined that there was no obvious risk of publication and that it was worthy of publication; therefore, I approved the manuscript for publication.